# Pervasiveness of Microprotein Function Amongst *Drosophila* Small Open Reading Frames (SMORFS)

**DOI:** 10.3390/cells13242090

**Published:** 2024-12-18

**Authors:** Ana Isabel Platero, Jose Ignacio Pueyo, Sarah Anne Bishop, Emile Gerard Magny, Juan Pablo Couso

**Affiliations:** 1Centro Andaluz de Biologia del Desarrollo, Universidad Pablo de Olavide, CSIC, 41013 Sevilla, Spain; aiplagom@upo.es (A.I.P.); sabis@upo.es (S.A.B.); 2Brighton and Sussex Medical School, University of Sussex, Falmer, Brighton BN1 9PS, UK; j.i.pueyo-marques@sussex.ac.uk; 3Department of Molecular & Cellular Physiology and Institute of Stem Cell Biology and Regenerative Medicine, Stanford University School of Medicine, Stanford, CA 94035, USA

**Keywords:** smORFs (small open reading frames), microproteins, sCDS (short coding sequences), *Drosophila melanogaster*, embryogenesis, proteomics, ribosome profiling, functional genomics, autophagy regulation

## Abstract

Small Open Reading Frames (smORFs) of less than 100 codons remain mostly uncharacterised. About a thousand smORFs per genome encode peptides and microproteins about 70–80 aa long, often containing recognisable protein structures and markers of translation, and these are referred to as short Coding Sequences (sCDSs). The characterisation of individual sCDSs has provided examples of smORFs’ function and conservation, but we cannot infer the functionality of all other metazoan smORFs from these. sCDS function has been characterised at a genome-wide scale in yeast and bacteria, showing that hundreds can produce a phenotype, but attempts in metazoans have been less successful. Either most sCDSs are not functional, or classic experimental techniques do not work with smORFs due to their shortness. Here, we combine extensive proteomics with bioinformatics and genetics in order to detect and corroborate sCDS function in *Drosophila*. Our studies nearly double the number of sCDSs with detected peptides and microproteins and an experimentally corroborated function. Finally, we observe a correlation between proven sCDS protein function and bioinformatic markers such as conservation and GC content. Our results support that sCDSs peptides and microproteins act as membrane-related regulators of canonical proteins, regulators whose functions are best understood at the cellular level, and whose mutants produce little, if any, overt morphological phenotypes.

## 1. Introduction

Small Open Reading Frames (smORFs) of less than 100 codons number in the hundreds of thousands in fruit flies (*Drosophila melanogaster*) and millions in the human genome, but remain mostly uncharacterised. According to their RNA features and aa sequence, smORFs belong to distinct classes, which correlate with their general function, from inert DNA sequences to transcribed ones, and finally, to sequences producing peptides that can regulate canonical proteins [1] (Figure 1). Thus, most smORFs appear dormant in the genome as non-transcribed intergenic ORFs with an average size of 22 codons, whose distributions, in terms of size and number, suggest a random origin. Around 10% of smORFs (tens of thousands) appear in RNA messengers (uORFs within the 5′ untranslated region of mRNAs, dORFs in 3′ UTRs, or lncORFs within putatively non-coding RNAs) and are therefore transcribed, but their translation is seldom detected [2]. Fewer still (1%, about a thousand per genome) appear in their own mRNAs, encoding peptides and short proteins of about 70–80 aa long, which are usually annotated as translated (Short Coding Sequences (sCDSs) [1]). These peptide sequences often contain recognisable protein structures such as transmembrane alpha-helices, which are present in 32% of all annotated sCDSs [3]. When characterised, sCDSs usually show detectable protein activity, producing peptides that are able to regulate and interact with canonical proteins [1]. Indeed, some of these sCDS can arise as shortened isoforms of canonical proteins, which work in a kind of ‘dominant-negative’ manner [4,5] (Figure 1).

smORF protein function has been difficult to ascertain. To date, more than 30 novel lncORF microproteins [2,6,7,8,9,10,11,12,13,14] have been characterised within misannotated non-coding RNAs (many from humans) and shown to have important functions in development, cell physiology, and metabolism, clearly establishing the importance of smORFs as a functional gene class. How many other translated smORFs remain unidentified? Recently, the development of ribosomal profiling has provided a tool to detect translated sequences at a genomic scale [15,16]. In *Drosophila*, Ribo-seq results [3,17] reveal the translation of thousands of smORFs, backed-up by similar data from vertebrates [15,18,19]. However, smORF translation does not automatically mean function. In some cases, uORFs and lncORFs have been shown to regulate their encoding transcripts through the act of translation, with the encoded microprotein being a side product without apparent function [7,20,21]. Further, smORF classes may represent stages in the evolution of de novo protein coding sequences from non-coding DNA [1,22]. From this perspective, many smORFs can be seen as protogenes [23] that have yet to acquire a full protein function.

Attempts have been made to characterise smORF function at a genome-wide scale in yeast and bacteria, showing that hundreds can produce a phenotype [24,25] and thus refuting the null hypothesis that “most smORFs are irrelevant and non-functional”. In metazoans, this null hypothesis was refuted through the discovery of individual genes, starting with the characterisation of the *tarsal-less (tal)* gene in 2007 [26]: *tal* is polycistronic and produces the shortest directly encoded bioactive peptides to date (11 aa); these peptides are essential for development and are conserved across arthropods, where they postranscriptionally regulate the Shavenbaby transcription factor [27,28]. In 2013, we characterised the *sarcolamban* smORF family, which includes the shortest smORF characterised in humans (31 aa *sarcolipin*). Remarkably, these smORF peptides control Ca2^+^-mediated heart rhythmicity from *Drosophila* to humans, and thus reveal not only an astonishingly wide functional conservation of smORF peptide function, but also the biomedical relevance of studying them in *Drosophila* [9]. These are not isolated cases; we characterised another smORF (88 aa) conserved from *Drosophila* to humans, which has conserved functions in endo-lysosomal maturation and macrophage phagocytosis [29], and a few other functional smORFs were characterised in *Drosophila* and other metazoans [7,30,31]. However, although these cases establish the important principles of smORF function and conservation, we cannot infer the functionality of all other metazoan smORF sequences from them. For a start, their high numbers contrast with their previous low rate of detection in biochemical and genetic experiments, which were focused on canonical gene and protein detection and characterisation [32]. Either numerous smORFs are functional but classic experimental techniques do not work with them, or most smORFs are not functional. Clearly, a smORF-specific effort is required to test the functionality of metazoan smORFs using experimental and bioinformatic means. An extensive effort to characterise human uORFs and lncORFs was recently carried out using cell lines [33], but the results must undergo ultimate validation in animal model systems.

Recently, a screening for smORF mutant phenotypes was carried out in *Drosophila melanogaster*, focusing on deeply conserved smORFs from *Drosophila* to humans [34]. Surprisingly, given that deep conservation is assumed to indicate function, only a small number of smORFs were observed to produce a phenotype when mutated, pertaining to either fly morphology or viability. A possible explanation for this is that the mutagenic method used did not cause strong enough losses of function. Aiming for high throughput, the authors used somatic CRISPR mutations, generating mosaic flies composed of mutant and wild-type cells. There is the possibility that not enough mutant cells were generated, or that the phenotypic focus was not affected.

Another source of dissonance is proteomics detection. Proteomics only detects about 40% of canonical proteins [35], but the proportion of detected smORFs is even lower [36]. An obvious explanation for this is the shortness of smORF proteins and peptides, which affects their stability and generates fewer trypsinated peptides for detection, as well as their widespread low levels of translation (and hence peptide production), as detected by Ribo-seq. Nevertheless, reliable peptide and protein detection using proteomics is an indicator of smORF protein production and stability, and hence is not only a corroborator of smORF expression but also a strong indicator of putative protein function (beyond Ribo-seq, which detects only the act of translation and is blind to the fate or function of the translated peptide).

Finally, smORFs’ shortness also defeats bioinformatics, which is the most-used and easiest way to identify sequence functionality, either through similarities to previously identified functional sequences or on the basis of extensive conservation across species. The presence of known protein domains whose biochemical function was characterised in canonical proteins is a strong indicator of function, but even a simple ORF and aa sequence conservation is an indicator of coding and protein function. However, given the shortness of smORFs, pure conservation can occur due to chance, and more detailed indicators (such as Dn/Ds or PhyloCSF) may lack enough signal to resolve this point in individual smORFs. At the genome level, given the low frequency (1%) of smORF translation, coding markers can be obscured by the non-coding smORF majority. We recently introduced a computational pipeline (GENOR) [37] to detect the homology of short protein sequences at both the individual and genome levels. When applied to smORFs, GENOR outperforms standard tools such as BlastP 2.2.25, which is known to struggle with microproteins [38]. Crucially, combining GENOR’s results with Ribo-seq reveals a positive relationship between smORF translation and conservation, with the reproducibly translated smORFs showing a longer conservation (gene age) and negative Dn/Ds values, indicating a purifying selection. An alternative approach is to use surrogate indicators of coding function (such as GC content [39,40,41,42]) that use the whole smORF nucleotide sequence, multiplying the signal up to three times with respect to aa. Nonetheless, to fully corroborate that a peptide is functional, and to characterize such function at the cellular and organism level, one needs to apply functional genetics.

Here, we combine an extensive peptidomic search with the bioinformatics indicators of function and with genetic experiments in order to detect and corroborate smORF peptide function in *Drosophila*. We focused on sCDSs since, as a smORF class, they have the best smORF markers of coding function (length; aa usage; the presence of structured domains; strength of Ribo-seq signal), with values approaching those of canonical ORFs. We also focused on function during embryogenesis, the life stage with the best and more complete genomic and gene expression data in *Drosophila*. Our studies increased the number of sCDS peptides and microproteins detected from 31% to 45% of those predicted by Ribo-seq and, together with previous data, we were able to experimentally prove a function for 63% of the sCDSs tested. Finally, we could observe a correlation between proven sCDS protein function and bioinformatic markers such as deep conservation across metazoans and GC content. Our results reinforce the pervasiveness and functional importance of smORFs and the microproteins they encode.

## 2. Materials and Methods

### 2.1. Genetics

#### 2.1.1. Hemizygous Tests

The viability of mutant smORFs was studied in a cross (♀ smORF-insertion /Balancer1 × ♂ Df (smORF)/Balancer2) and scored as follows: number of Insertion/Df flies relative to Insertion/Balancer siblings. The crosses were always carried out with Df carried on the males to minimize the possible, smORF-unrelated, maternal effects of the other genes made hemizygous by the Deficiencies. Whenever possible, smORF mutants were independently tested against different Deficiencies, and Deficiencies covering the same locus were tested for lethality against each other.

#### 2.1.2. Knight–Robertson Tests

We used the Knight–Robertson balancer test method [43], where a viable smORF insertion under assessment is introduced in a small population in competition with Balancer chromosomes, initially in 1:1 allelic frequencies. The population was then left to breed for two or three generations, and the flies were screened and genotypes counted after each generation. Then, the ratio of observed Insertion/Insertion flies versus Bal/Bal was calculated. Since Balancer chromosomes are homozygous lethal, their relative allelic frequency should drop from 0.5 to 0.25 in two generations when competing with wild-type chromosomes (even without taking into account the deleterious effects of the Balancers’ dominant mutations and of their many recessive markers and chromosomal breakpoints). Thus, an Insertion/Bal ratio 2–3 generations below 1 indicates that the smORF insertion reduces fitness even more than Balancers, whereas a ratio of 1 indicates that the tested insertion is as deleterious as a Balancer, and hence, is also less fit than the wild-type.

#### 2.1.3. Generation of CG34250 Null Mutants

This was achieved by crossing parental strains containing two different deletions that overlap on the *CG34250* gene locus to generate transheterozygous offspring missing the *CG34250* gene. The R12.2 deletion affected part of the *CG7580* gene and the entire *CG34250* gene and was generated by an imprecise excision of the P{SUP o-P}CG7580[KG01913], triggered by a Δ2–3 transposase source [44]. The other deletion was generated by inducing a precise FRT recombination between the Pbac{WH}f01614 and Pbac{WH}qjt[f01023] using Flipase [45]. This deletion depleted the 5′ end of the *qjt* gene and the entire *CG13733* and *CG34250* genes. The deletion mapping of both deletions was achieved by PCR amplification using specific primers of the *CG34250* and nearby genes and sequencing.

#### 2.1.4. Generation of the CG34250 Ct Tagged Venus and CG12384 Ct Tagged FLAG Constructs

The 5′UTR and ORF of the smORF were amplified by PCR using the following primers: CG34250F 5′CACCGCACAGCTGTTTGCAAGCAAAATA3′; CG34250R 5′CGTGACAGAAAAGCTGCGAAAAT3′ and CG12384F 5′CACCATTCGATAGACAACAGTCGGACG3′; CG12383R 5′CTTCCGTGGCTGCTGGATGTG3′. PCR products were cloned in the pENTR™ Directional TOPO^®^ Cloning vector (Invitrogen; Thermo Fisher Scientific Inc., Waltham, MA, USA). The integration of smORF fragments to final destination vectors (https://carnegiescience.edu/bse/drosophila-gateway-vector-collection obtained from the Drosophila Genomics Research Center) was achieved by LR recombination following the Gateway system (Invitrogen; Thermo Fisher Scientific Inc., Waltham, MA, USA).

#### 2.1.5. Reverse Transcription from Unfertilized Eggs

RNA was extracted from unfertilised eggs and reverse-transcribed to generate the cDNA pool using Invitrogen’s Moloney Murine Leukemia Virus Reverse Transcriptase Kit. PCR reactions were conducted using reagents provided in the Taq PCR Core Kit (QIAGEN, Hilden, Germany) using CG34250F 5′GCACAGCTGTTTGCAAGCAAAATA3′ and CG34250R 5′TTCAGCAGCATGACCACGAAGA3′ primers, amplifying a 197 bp product.

### 2.2. Bioinformatics

#### Homology Detection, Sequence Alignment, and Phylogenetic Trees

To identify homologues of Drosophila melanogaster sCDS, first Flybase and EMBL databases, and then the Appendix A of Bosch et al., 2023 [34], were consulted, either in bulk or with individual gene queries, and the results were collated. For the two genes characterised experimentally (*CG34250*, and *CG12384/dap1*), their aminoacid ORF sequences from Drosophila melanogaster were used to further search for homologues in other species, using PSI-BLAST (NCBI) to query protein databases. Specific smORF homologues were then selected for sequence alignment and phylogenetic tree. We used both the COBALT Multiple Alignment tool in NCBI [46] and Clustal 0 (1.2.4) in the Uniprot web platform. Clustal 0 was used for the multiple alignment of smORF sequences, highlighting similarities. The Neighbor Joining tree method was used for phylogenetic analysis with maximum sequence difference 0.852.2.2 GC content measurements. Data were obtained using the query tool from Flybase and then manually curated to increase their consistency. sCDS sequences were obtained by querying for ORFs of less than 100 codons; sCDS with an isoform longer than 100aa were removed, and repeated peptide isoforms (PB, PC, etc) with identical or 80% overlapping regions were also removed. lncORFs were extracted from annotated non-coding RNAs with embryo transcription, with a limit of five lncORFs per RNA. Pseudogene gene data were calculated from pseudogene exons. For canonical genes, a random sample of ORFs from named (characterised) genes was obtained. GC content was calculated as the GC ratio (G + C/G + C + A + T).

### 2.3. Tissue Culture

*Drosophila* S2 cells (S2-DRSC, Stock 181) were sourced from the Drosophila Genomics Resource Centre (Indiana, USA). The cells were grown at 25°C under standard conditions in Schneider’s medium with 10% FBS and 1% penicillin/streptavidin. Transfection was carried out as described in Pueyo *et al.*, 2016 [29] using pUASt-smORF-tagged and Act5Gal4 vectors. A total of 1 mL of S2 cells was placed on top of acid-treated coverslips and left to adhere for 1hr; then cells were fixed with 4% paraformaldehyde for 15 min and washed several times with PBT (PBS + Triton X-100 at 0.2%) for 30 min. Cells were blocked with 2% BSA in PBT for 30 min. The primary antibodies mouse FLAG-antibody (1:500—Sigma–Aldrich, St. Louis, MO, USA) and rabbit polyclonal Ref(2)P antibody (1:1000 gift by Erik Rusten) were incubated overnight at 4 °C. After several washes with PBT and cells were incubated with the secondary antibodies anti-rabbit FITC (1:500—Jackson Immunoresearch, West Grove, PA, USA) and anti-mouse-Rhodamine (1:400—Jackson Immunoresearch) for 1hr. After several PBT washes for 30 min, cells were washed in PBS and mounted in antifade mounting media with DAPI (Vectashield, Vector, Newark, CA, USA).

### 2.4. Proteomics

#### 2.4.1. PunchP

We performed 29 PunchP experiments using *Drosophila* S2 cells, following the protocol described by Aviner et al., 2013 [47]. This involved the incorporation of puromycin into nascent peptides as they were released from the ribosome, allowing us to analyse protein synthesis dynamics in these cells. The experimental setup closely followed the methodology detailed in the original publication, with slight adjustments: we incorporated a size-exclusion column to enrich the sample in small peptides up to 10 kDa, which is roughly equivalent to about 100 aa (see Appendix A). A size exclusion chromatography was performed with a Superdex 75 10/300 GL column (Cytiva, Marlborough, MA, USA) in an ÄKTA pureTM system (Cytiva). The separation of proteins was carried out at room temperature in a Urea-SDS buffer (Urea 8M, SDS 2%) at 0.75 mil/min, and fractions of 500 μL were collected and analysed.

#### 2.4.2. Gel Size-Fractionation

We performed gel size fractionation to separate biomolecules from Drosophila S2 cells (four experiments) and early-developmental-stage Drosophila embryos (0–8 h) (three experiments). These embryos were previously dechorionated to facilitate the extraction and analysis of intracellular components. After obtaining the protein gel (sodium dodecyl sulfate polyacrylamide gel electrophoresis, SDS-PAGE), we excised the bands corresponding to a size of approximately 10 kDa. The excised acrylamide gel bands were then distained with ammonium bicarbonate and acetonitrile. To reduce and alkylate the proteins, dithiothreitol (DTT) was used to break disulfide bonds, and iodoacetamide was applied to carbamidomethylate cysteine residues. The samples were incubated overnight at 37 °C with bovine trypsin (Promega, Madison, WI, USA) at a 1:10 enzyme-to-substrate ratio. Following extraction with acetonitrile and acidification, the samples were desalted and concentrated using C18-filled tips (OMIX, Agilent Technologies, Santa Clara, CA, USA).

#### 2.4.3. Mass Spectrometry Analysis

Protein digested samples were dried, resuspended in formic acid 0.1%, and separated in a Thermo Scientific Easy nLC system using a 50 cm C18 Thermo Scientific (Waltham, MA, USA) EASY-Spray™ column. The following solvents were employed as mobile phases: water 0.1% formic acid (phase A) and acetonitrile, 20% H2O, 0.1% formic acid (phase B). Separation was achieved with an acetonitrile gradient from 10% to 35% over 240 min, 35% to 100% over 1 min, and 100% B over 5 min at a flow rate of 200 nl/min.

A Thermo Scientific^TM^ (Waltham, MA, USA) Q Exactive™ Plus Orbitrap™ mass spectrometer was used to acquire the top 10 MS/MS spectra in DDA mode. LC-MS data were analysed using the SEQUEST^®^ HT and MASCOT search engine in Thermo Scientific™ Proteome Discoverer™ 2.2 software using static carbamidomethylation (C), and dynamic oxidation (M) modifications. Data were searched against our own protein database with 4000 entries corresponding to our previous positives and borderline results for smORFs with Ribo-seq with a footprint RPKM > 0 [36,37]) because we verified that searching MS data against databases that are too large can increase the possibility of false discoveries. The results were filtered using a 1% protein FDR threshold to reduce misidentifications. Data analysis was performed with all peptides identified in the different PunchP and Gel experiments that were determined to be high confidence by the Mascot search engine. The data were later transferred to Excel for statistical analysis (see Appendix A).

### 2.5. CG12384/DAP Lysosomal Stainings

CG12384^LL30852^ (y* w*; P{neoFRT}40A P{FRT(whs)}G13 cn1 PBac{SAstopDsRed}LL03852 bw1) flies were reared in in standard cornmeal media; third instar homozygous or heterozygous larvae were either kept in standard media (fed) or transferred to empty tubes with humid paper towels for three hours (starved). Female larvae were then everted in ice-cold PBS, incubated in 50 nM Lysotracker-red (DND-99) (1:1000; Invitrogen; Thermo Fisher Scientific Inc., Waltham, MA, USA)) for 30 min at room temperature, rinsed twice in PBS, fixed in 4% paraformaldehyde in PBS for 20 min at room temperature, and washed three times in PBS with 0.3% Triton-X-100, under rotation (10 min per wash, at room temperature). The everted larvae were then incubated in 1 µg/mL DAPI, in PBS, for 20 min, and then rinsed once more in PBS. The fat bodies were then separated from the rest of the larvae and mounted on a microscope slide within a Secure Seal imaging spacer adhesive well in Vectashield, and imaged with a Zeiss laser scanning microscope LSM 5.10 on a Zeiss Axioskop 2 stage, with a 40× Achroplan objective. The lysotracker intensity was quantified using Image J 1.54f; for each fat body, we considered the average of two independent ROIs, each covering approximately half of the imaged fat body. S2 cell transfection and antibody stainings were performed as described by Patraquim et al., 2022 [37].

## 3. Results

### 3.1. Proteomic Detection of smORF Peptides

Proteomics is used to detect peptide production, and hence translation and ORF coding function. However, proteomic detection requires two factors: a high protein production rate, and high protein stability. The most exhaustive studies in *Drosophila* only detect 60% of canonical proteins [35,36], preferentially those with high translation levels [36]; furthermore, proteomics especially fails to detect small proteins and peptides (presumably due to their lower stability, possibly a result of an intrinsically disordered structure and reduced resistance to proteases) [18,36]. We targeted proteomics to annotated sCDS proteins using two different techniques. First, we tried gel size-fractionation [3]. Second, we applied PunchP [47] to detect smORF nascent peptides (translating peptides still attached to the Ribosomes, Appendix A) by introducing the step of size-fractionation (see the Section 2 and Appendix A). We applied these techniques to both embryos and S2 cell cultures, which are 98.5% overlapping in the translation of sCDS smORFs (as befits the S2 cell line being derived from embryos [48]). Comparing our proteomics with previous *Drosophila* embryonic and S2 cell data obtained in extensive proteomic screenings [35,36] shows an improvement in detection rate, as we detected 182 sCDS peptides, including 88 not detected by previous efforts (Figure 2A; Appendix A). Another 16 embryo-translated sCDS peptides were previously reported but were not detected by us, yielding a total 47.5% overlap (94 detected here and previously vs. 104 detected in either but not both; Figure 2A). A comparison with previous Ribo-seq results [37] shows that we detected 182 out of 357 (51%) sCDS peptides translated in embryos, eggs, or S2 cells, and that, in total, 55.4% of the sCDS proteins (198 out of 357) expressed in these tissues have now been detected by proteomics (Figure 2B). In addition, our proteomics detected an extra 18 peptides that might be proteomics misidentifications or misclassified by Ribo-seq. This is probably the case for six proteins: two were also detected in the previous proteomics literature; two had an ambiguous Ribo-seq signal; and another two had a clear RNA-seq signal, leaving thirteen possible false positives. This would represent a false positive rate of 7.1%, indicating that our methods can reliably detect sCDS peptides of around 80aa, including many below this size (Appendix A). A comparison of the Gel fractionation with the PunchP results (Appendix A) does not reveal any major difference in the size of the sCDS proteins detected (medians of 82 vs. 83 aa, respectively), with both being slightly skewed towards longer sCDSs, with an overall median of 77aa. Gel fractionation experiments are more productive in detecting sCDS (Appendix A). This is probably due to the number of canonical proteins (longer than 100aa) that are co-detected: given that a typical proteomic experiment can detect a finite number of peptides overall, the greater the number of canonical detections, the fewer the sCDSs. We introduced a size-fractionation in Punch-P, but this would not remove nascent canonical proteins that still had not elongated beyond 100aa. Thus, a typical PunchP experiment detected 2.55 sCDS microproteins at a ratio of 0.02 sCDS for each canonical detection, whereas a Gel fractionation experiment detected 23.7 sCDSs at a ratio of 0.09 sCDS microproteins for each canonical (Appendix A). Overall, our proteomic approach doubled the number of sCDS peptides detected in embryos, corroborating almost half of those that appeared to be translated by Ribo-seq [36]. This corroborated proteomic detection supports the strong translation of stable sCDS peptides, which is a prerequisite for their biological function.

### 3.2. Bioinformatics Indicators of Function

#### 3.2.1. GC Content

The fraction of GC versus AT nucleotides has long been regarded as an indicator of coding potential in many organisms, including humans and *Drosophila* [39,40]. Here, we observed (Figure 3A, Appendix A) that, in *Drosophila*, GC content varies from an average of 0.40 in intergenic (untranscribed) ORFs to an average of 0.54 in canonical ORF proteins (longer than 100 codons). The average GC content of sCDS ORFs in *Drosophila* is 0.51, while the average for lncORFs (smORFs in lncRNAs) is 0.47, which is very similar to that of pseudogenes, at 0.48. Thus, the GC content fits with the observations that only report other markers of productive translation in a minority of lncORFs and pseudogenes [50]. Further, the GC content variation correlates with their average lengths (Figure 3B), a feature that has been related to translation potential and smORF age [37]. Interestingly, the GC content of the ORFs within the *tarsal-less* gene is 0.57. This gene resembles a translated lncRNA because of its short ORFs of 11 to 32 codons arranged in polycistronic transcripts, (and was, in fact, annotated as a lncRNA [51] before its translation and peptide function was proven [26]), but their GC content reveals their coding function. Thus, the GC content suggests that, on average, *Drosophila* sCDSs have a coding function; that is, they are translated into functional peptides and small proteins.

#### 3.2.2. smORF Conservation

Sequence conservation across species, and hence across evolutionary time, is one of the best indicators of function. We studied sCDS amino acid sequence conservation at two levels of depth: (1) in other Drosophilids (*Drosophila* genus species, diverged <50 million years ago, -Mya) and (2) in humans (>500 Mya). For Drosophilids, we used our pipeline GENOR [37], which we have shown to outperform genome annotations and standard engines such as BlastP. For human conservation, we used annotated data from Flybase, EMBL, and Bosch et al., 2023 [1,34]. We observe that 33.6% sCDSs (298 out of 887 in *Drosophila melanogaster* [36]) have a potential homologue in humans, whereas 47.5% have homologues in other Drosophilids. Interestingly, only 20% of these Drosophilid-conserved sCDSs (i.e., less than 10% of the total) were found to be conserved in more distant insects and arthropods. These data could be explained by the paucity of well-annotated insect genomes, but prima facie might indicate that sCDSs in *Drosophila* underwent two rounds of expansion, one before 500 Mya, together with other metazoans, and another at about 200 Mya, following the dipteran radiation. This generative process seems to continue, as at least nine sCDSs did not have any homologues, suggesting that they are novel genes that arose in *Drosophila melanogaster* in the last 5–20 Mya. Notably, 60% of the human-conserved sCDSs were still cognate gene predictions (CGs) that is, uncharacterised genes, without a given name and proven function, whether via one-to-one homology, or genetics and other experimental assays; most of the dipteran-conserved sCDSs were also of this CG kind (see below). This paucity of smORF gene characterisation (which, amongst canonical *Drosophila* proteins, is limited to about 24% of the ORF-ome) could again be explained by their shortness, which complicates homology detection and also presents smaller, less detectable targets for random genetic screens.

### 3.3. Genetic Requirements for smORF Genes

#### 3.3.1. To Investigate the Functionality of Uncharacterised, ‘CG’ sCDSs, We Carried Out Genetic Tests

smORFs have revealed a variety of organismal requirements, often acting as regulators of canonical proteins [1,7]. Functional tests for regulators such as smORFs need to be wide and unbiased, avoiding a focus on overt phenotypes such as an abnormal morphology. In principle, the loss of function of a regulator increases the chance of the production of a subtler or weaker phenotype than the loss of function of the protein being regulated. Thus, we designed a pipeline for the genetic testing of mutant smORF viability. We collected publicly available transposon-derived transgenic insertions in a sample of sCDS transcripts showing embryo expression (in embryonic, egg, or S2 cell Ribo-seq samples), without described mutant phenotypes. We also collected deletions and deficiencies covering the affected sCDS genes (Appendix A). Altogether, we tested mutants for 43 sCDSs, and observed lethality, lower viability, or lower fitness in 20 (46.5%) of them (see below; Table 1).

#### 3.3.2. Hemizygous Genetic Test

First, we scored the viability of the sCDS insertions under hemizygous conditions (i.e., over a Deficiency covering the locus), relative to sibling controls, and scored them as low if below 0.7 of controls, semi-lethal if below 0.1, and lethal if 0 (see Methods, Appendix A, and Table 1). We tested several alleles (deficiencies and mutant insertions per sCDS transcript) to minimize the confounding effects of either particular transgenes and deficiencies or of chromosomal modifiers (undescribed, yet potentially interacting mutations in each chromosome). Notice that our design involves the pairing of two chromosomes (insertion and deletion) of independent origins, which are thus unlikely to contain the same modifiers. Second, this design should produce a higher loss of function than that produced by hypomorphic alleles in homozygous conditions, and thus be more likely to reveal a mutant phenotype. Generally, we observed few differences in the results while using different alleles. When in conflict, we gave preference to results produced with a smaller deficiency and with insertions inside the ORF or otherwise truncating it (such as LL insertions; see below for those of the *CG12384* gene); these were given more weight than insertions elsewhere in the transcript. These tests uncovered a requirement for 18 sCDSs and showed apparently total viability for another 25. We only observed a morphological phenotype with two genes (*CG17278* and *CG7646*) (see Table 1).

#### 3.3.3. Fitness Tests

We reasoned that some requirements may have escaped the hemizygous tests, particularly those involving maternal or multigenerational components or those requiring a higher loss of function of the assessed gene to become evident. Thus, we carried out a further test with those sCDS mutants that did not show lowered viability in hemizygous conditions (plus two controls that had). Fitness has long been used in population genetics, but recently also emerged as a sensitive detector of gene function [52,53]. We used the Knight–Robertson (K-R) balancer test method, where the transgenic insertion under assessment is introduced in a small population in competition with balancer chromosomes, initially at 1:1 allelic frequencies. The population is then left to breed for two or three generations (see [43] and Section 2). After this period, the ratio of allelic frequencies reveals the fitness of the P insertion versus the balancers. Because balancers are homozygous lethal and carry numerous mutations and breakpoints, their frequencies are expected to decay rapidly, so that even the maintenance of the original ratio of 1 P/Balancer is indicative of a fitness problem. As this test is carried out over several generations, the deleterious effects accumulate and increase, revealing any problems with fecundity as well as viability. We carried out these tests with 16 sCDSs, in some cases testing several insertions. These fitness tests corroborated the requirements shown by the two control sCDSs and uncovered the requirements for another three. For example, the gene *CG7646* had an insertion in its ORF described as lethal, which we found to be viable over a deficiency for the locus. However, we observed that 50% of the resulting flies showed wings that were held up at right angles to the body. This held-up phenotype is associated with defects in the neuromuscular flight complex, and should impact the behaviour of the fly, not only impacting its ability to fly but also impairing the males’ ability to court females (since the males need to “buzz” or “sing” to females, with their wings beating at high speeds). Although we scored *CG7646* as having this mutant phenotype in the hemizygous test, the Knight–Robertson test also showed that the Balancer frequency did not drop, corroborating the suggestion of a fitness problem (Table 1).

Our genetic tests uncovered a requirement, and hence, a function, for 21 out of 46 sCDSs tested, a 45.6% fraction.

#### 3.3.4. In Situ Expression

In order to understand the functional focus of the uncharacterised sCDSs, we carried out an *in situ* hybridisation in *Drosophila* embryos with a sample of the assessed sCDSs (Figure 4) and compared and completed our data with the information available from the Berkeley *Drosophila* Genome Project (Appendix A). We observed that most patterns of expression of the sCDSs that we assessed genetically were either initially ubiquitous (possibly reflecting a maternal expression), or were present in mesodermal and endodermal tissues before resolving to specific organs. These results suggest that many sCDSs may be acting in the development and/or the physiological function of internal organs and the cells therein, and a detailed functional characterisation of those organs would be required in order to detect their function; otherwise, they would be classified as apparently non-functional. This observation is compatible with the paucity of overt morphological phenotypes, both in our study and in the work of Bosch et al., 2023 [34], who also observed a similar clustering of sCDS expression in the muscles, guts, and other internal organs.

### 3.4. Validation of Genetic Results

Both the hemizygous viability and the K-R tests report “silent” or abstract gene requirements, and are not primarily designed to expose overt mutant phenotypes (although these can be observed on occasion). In order to validate our genetic results and to obtain more precise information on the function of sCDSs, here we characterised two sCDSs in more detail, adding to others characterised elsewhere [4,54].

#### 3.4.1. CG34250

We selected the *CG34250* gene, which produced a viable result in the hemizygous test (with a P/Df to P/Bal ratio of 1.25) but a reduced fitness in the K-R test, with a P/Bal ratio of 0.10 (see Appendix A). Importantly, the fitness of a wild-type P-element revertant strain in K-R test was reverted to wild-type levels, indicating that the fitness reduction was due to the P-element insertion in *CG34250* gene locus (Figure 5A; Appendix A). Thus, we expected this gene to have a subtle function. *CG34250* encodes a 54aa peptide that was also detected via proteomics in embryos (Figure 5B and Appendix A). This peptide belongs to the 32% of sCDSs containing a putative transmembrane alpha-helix domain [3] and is conserved across Drosophilids and other dipterans (Figure 5B–F) but apparently not in humans, and its ORF has a GC content of 0.50 (just below the average for sCDSs; see Figure 3, Appendix A). To discard the possibility that the studied insertions did not generate a sufficient loss of function, we created two overlapping deletions that, in heterozygosis, completely delete only this gene (Figure 5A; see Section 2). When the parental heterozygous for each deletion over the balancer crossed, they produced *CG34250* null progeny with normal viability and with no obvious morphological abnormalities. However, we noticed that *CG34250* was maternally expressed (Appendix A, Figure 5J), which we corroborated via an RT-PCR of unfertilized eggs derived from *CG34250* females (Figure 5I). Thus, we carried out an F2 viability test, where we scored the viability of *CG34250* mutants descended from *CG34250* null mothers. Surprisingly, we observed no effect until the crosses were carried out in such a way that the *CG34250* null progeny from *CG34250* null mothers were raised in competition with non-mutant siblings (Appendix A). This result might suggest a behavioural or physiological problem in *CG34250* mutants. We generated a carboxyl-terminus tagged version of the *CG34250* peptide and observed its expression in S2 cells. CG34250-Venus can initially be observed in a reticular and perinuclear distribution, perhaps reflecting its synthesis in the ER as a membrane peptide (Figure 5G–G’’’). When accumulated at higher levels, it can be seen in larger intracellular membrane compartments colocalizing with membrane marker cd8-mRFP and lysosomes (Figure 5H–H’’’), reflecting either its degradation, or its functional location. This second possibility is interesting when considering the *CG34250* mutant phenotype, given the physiological and physical alterations that the loss of lysosomal proteins produces in humans [55]. Altogether, the analysis of *CG34250* indicates that the functional requirements for even recalcitrant smORF genes can be exposed through adequate testing once information about their possible functional focus is observed either through gene expression tests or bioinformatics markers.

#### 3.4.2. Human-Conserved sCDS Involved in Autophagy

We also reasoned that some sCDSs may have evaded our screen, since both viability and fitness may vary under different conditions and we were only assessing them under standard ones. Thus, we selected an apparently fully viable sCDS, which had otherwise good functional markers, for further characterisation. We selected *CG12384* for further study, as the possibly null insertions in this gene did not reduce its viability (Table 1 and Appendix A), yet a 96 aa protein was detected using our proteomics (Appendix A) and its sequence showed a high GC content of 0.62 and a potential human homology. This sCDS encodes a potential homologue of human death-associated protein 1 (DAP1) with homologues found throughout animal phylogeny from arthropods to vertebrates (Figure 6A,B). DAP1 has been implicated in TOR signalling as a negative modulator of autophagy [56] (Figure 6C,D) and we noted that Flybase reported high expression in the fat body, which acts as a nutritional reservoir for flies. We observed full viability with a transgenic insertion in the 5′ UTR of this transcript outside the ORF (P(SUPor-p)KG03371), but also observed that an intronic insertion carrying a mutagenic gene trap cassette that inserts an artificial exon with stop codons (PBac{SAstopDsRed}LL30852), and hence truncates the protein sequence, was also fully viable over a deficiency. This insertion allowed us to recover homozygous mutant larvae, in which we observed increased autophagy under starvation, whereas the overexpression of this peptide prevents this effect. This effect is noticeable in the fat body, where we could observe larger lysotracker-labelled vesicles in CG12384^LL30852^ homozygous mutants compared to heterozygous controls (Figure 6E–I). We also observed a similar effect in S2 cell cultures. Under starvation (serum-free culture), S2 cells showed a high expression of the autophagosome marker Ref(2)P (J), but this expression was reduced or completely abolished in cells constitutively expressing a CG12384-Flag-tagged peptide (Figure 6K) under the expression of a constitutive promoter (Act5). We note that, together, these results are compatible with a function for CG12384 as a negative regulator of the autophagosome downstream of TOR, as suggested for DAP1. Thus, our results support the homology of CG12384 with DAP1, both at sequence (Figure 6A,B) and functional levels, and corroborate the proposed function of this new gene family as a negative regulator of autophagy (Figure 6E–K). We named the *Drosophila* gene *dap* (*defective autophagy)*. The *dap/CG12384* function supports our conclusion, as can be seen in genetic and *in situ* hybridisation data showing that sCDSs have a tendency to act in internal organs as regulators of cellular and/or physiological processes, and that this tendency may mask their requirements.

### 3.5. Comparison of Functional Indicators

We compared our results with the recent screening by Bosch et al., 2023 [34]. In their screening, the authors used CRISPR and screened, similar to us, for reductions in mutant viability under different conditions. They screened 130 smORFs conserved from flies to humans using somatic CRISPR, but observed mutant phenotypes in only 2.6% of the cases (36 sCDSs). This could be explained by the incomplete mosaicism of somatic CRISPR mutagenesis. However, these authors also generated whole-animal KOs for 27 sCDSs (which, in principle, are comparable to our 46 insertional mutants), and when using those KOs, the frequency of mutant phenotypes rose 36.5%, similar to the 45.6% that we observed. In total, our work and that of Bosch et al. overlap for 16 sCDSs, and both yielded the same result in nine cases (three mutant and six wild-type phenotypes), whereas Bosch et al., 2023 [34] (but not us) observed mutant phenotypes in two cases, and this work (but not that of Bosch et al., 2023 [34]) observed a phenotype in five sCDSs. Interestingly, in both cases, only a few examples of morphological defects were observed (two in this work and two in that of Bosch et al., 2023 [34]). We conclude that our results, and those of Bosch et al., showing either a somatic mutant phenotype or involving whole KO mutants, can be pooled together (for a total of 76 sCDS; see Appendix A) and compared with the functional indicators shown above (proteomics and bioinformatics).

We obtained putative functional indicators for the sCDS in the *Drosophila melanogaster* genome [36] and compared these with the 76 sCDSs that were tested for function using genetic tests by either us (Table 1) or Bosch et al., 2023 [34], as shown above. Proteomic detection revealed a relationship with genetic requirements, as the fraction of sCDS detected was higher (62% vs. 55%, Table 2) amongst those with a mutant phenotype and those without. There was also a difference in GC content, which was slightly higher for both groups than the average for all sCDS, but sCDSs with the mutant phenotype matched the GC content of canonical proteins (0.54%, see Figure 3). Drosophilid and human conservation was also higher in sCDSs with a mutant phenotype, but this could be due to the fact that Bosch et al. selected their pool because they appeared to be conserved in humans; however, amongst our own pool of 46 studied sCDS, human conservation was also higher for sCDS showing a mutant phenotype (42% vs. 25%). There was also a higher percentage of RNAi phenotypes reported in the literature amongst the ‘mutant’ group (Table 2; see also [54]). In the synthesis, we observed higher values for most putative functional markers amongst those sCDSs showing a genetic requirement. Whereas, individually, the differences are slight, the accumulation of higher values for several of these markers seems to be a good indicator of function. Altogether, the data indicate a functionality of up to 62% (47 out of 76) of the assessed sCDSs.

## 4. Discussion

smORFs are increasingly being recognised as an important, yet not fully characterised, part of our genomes. Many smORFs that produce peptides and microproteins with important biological functions have been characterised [1,7,8], but the number of uncharacterised smORFs is still several orders of magnitude higher. This is first because of their sheer number—they outnumber canonical ORFs by at least 10-fold—and because the shortness of smORFs challenges our standard gene characterisation tools. These two factors generate a ‘catch-22’ effect, whereby smORFs are generally considered non-functional (since so few have been characterised in depth), while individual smORFs are not chosen for in-depth characterisation (because they are considered non-functional at large). In order to break this impasse, we need high- and medium-throughput tools to screen for smORF functions at the genome level, and then to follow up and validate the results with a detailed characterisation of individual examples.

smORFs appear in different classes; amongst those, sCDSs most consistently contain functional smORFs (Figure 1) [1,4,8,54,57]. This empirical and anecdotal observation is backed up by the pervasive presence of functional markers amongst sCDSs. Hence, it makes sense to first characterise this smORF class before we embark on the genome-wide characterisation of more challenging smORFs classes, such as lncORFs, uORFs, and dORFs. We carried out a screening for sCDS function in *Drosophila* using a combination of biochemical, bioinformatic, developmental, and genetic techniques. The combination of these techniques can pinpoint smORFs with a coding function, producing biologically detectable and active peptides and microproteins. The results can then be used to identify promising smORF candidates for detailed characterization [4,54].

### 4.1. Genome-Wide Evidence for smORF Function at the Coding and Protein Levels

Ribo-seq is perhaps the most sensitive indicator of smORF translation, and can be used at the genome level as a first test of smORF functionality. However, translation is not synonymous with peptide and microprotein function [20]. Ribo-seq data can be combined with proteomics; however, as shown, the proteomics data for sCDSs are still not exhaustive, and when proteomics is applied to lncORFs and uORFs, it can produce results that are at odds with Ribo-seq (Platero and Couso, unp. obs.). Further, the absence of Ribo-seq signal in a given organ or tissue at a particular life stage does not exclude the possibility that the smORF in question may be translated in another place or at a different time. However, we now have a near-exhaustive sampling of smORF translation via Ribo-seq during *Drosophila* embryogenesis [36,37], such that we can narrow the search for smORF functionality to the detected translated ones.

Proteomics is not a full gene-discovery tool, since it basically tests the presence of predetermined sequences in an ORF-ome by matching observed mass-spectrometry spectra with theoretical predictions based on the ORF-ome sequences. This is contrary to, for example, RNAseq (and even Ribo-seq), which can detect RNA sequences de novo and then assemble them in a transcriptome from which ORF-omes can be extracted. Furthermore, proteomics consistently fails to detect many smORF peptides and microproteins flagged as translated by Ribo-seq, and it remains an open question if this is because said peptides do not exist in detectable amounts or if there are proteomics-specific biases that preclude consistent small peptide detection. Nonetheless, proteomics can be adapted to peptide discovery (peptidomics), and the simultaneous detection of smORF peptides by both Ribo-seq and proteomics is a strong indicator of the abundant production of stable peptides and microproteins and, hence, a strong indicator of smORF function.

Using two peptidomic techniques, we detected peptides for 51% of the sCDSs expressed during *Drosophila* embryogenesis, increasing the total detected pool by previous authors from 31% to a combined total of 55%. Most of the previous embryonic proteomic sCDSs that were detected were also detected in our samples (Figure 2 and Appendix A), showing that our peptidomics techniques (particularly gel fractionation) are well-suited to peptide and microprotein detection. However, about 200 sCDSs remain undetected. Similar to what was observed for canonical proteins, a comparison with the Ribo-seq data [36] suggests a correlation between embryonic sCDS proteomic detection and the intensity of translation. This might indicate that organ- or even cell-specific efforts might be required to complete sCDS peptide detection at the genomic level. Given the slight relationship we found between proteomic detection and genetic requirements (Table 2), this may be a worthwhile enterprise.

Bioinformatics markers can identify the average differences between translated and untranslated smORF populations and can also similarly identify the average makers of functional peptide and protein activity [37,52] (Table 2). However, no single bioinformatic marker is a watertight smORF function indicator, and even the observation and combination of several markers can offer contradictory rather than synergistic results for a given gene. Hence, embarking on a detailed molecular and functional characterisation of a smORF gene solely based on bioinformatic data still entails a leap of faith.

### 4.2. Microprotein Function Can Frequently Be Detected Amongst Drosophila smORFs

One way to narrow down the candidates for a detailed characterisation and increase the odds of obtaining informative results is to carry out a genetics-based screening. Genetics reveals gene function using the phenotype produced by mutations (and analogous alterations), and a detailed observation of the phenotype can provide information not only regarding the presence of gene function but also its main characteristics. The information is more complete when coupled with other indicators, such as the pattern of gene expression. However, a screening to identify smORF gene function at a medium or large scale must be as unbiased, high-throughput, and widely applicable as possible. This ‘wide aiming’ screening range is required due to the possibility that smORFs may have few or subtle functions, as implied by their biochemical characteristics (short aa sizes that preclude both major structural functions and complex interactions based on multiple protein domains). These sequence characteristics suggesting subtle functions are often combined with a low signal, as detected by Ribo-seq and proteomics, suggesting low microprotein production levels. Thus, we designed a method based on two simple genetic tests, aiming to score the basic outputs of gene function, such as viability and fecundity. Aiming for simplicity, we carried out these tests using publicly available mutations, such as transgenic insertions and gene deficiencies. Using these tools and tests, we observed evidence of gene function in at least 46.5% of the smORFs that were screened (Table 1).

Our observations are in line with genome-wide screenings of smORF function in bacteria [25], yeast [24,52], and human cell lines [33] in that they show a high chance of detecting a functional requirement for smORF peptides and microproteins. However, both our work and that of these studies highlight the importance of targeting the screenings and employing the right screening techniques. Thus, Bosch et al., 2023 [34] also targeted sCDSs for genetic screening at a medium to large scale but were only able to identify function (based on loss of viability and/or fecundity) for a mere 2% of the screened genes. This is likely due to their use of CRISPR-mediated somatic mutations. In principle, CRISPR-mediated somatic mutations can be quickly obtained in a single generation (by crossing flies expressing a CRISPR sgRNA guide to others expressing Cas9 and screening the F1 progeny for phenotypes). It is an interesting idea to deploy a precise mutagenic agent in a large number of flies. However, the low rate of success suggests some technical or design problem occurred, most likely a lower-than-expected rate of mutagenesis, which produced mosaic individuals with a large enough fraction of unmutated cells to produce a suitable gene function and avert the presence of a mutant phenotype at the whole-animal level. This conclusion is corroborated by the much higher rate of mutant phenotypes observed when the authors used a multi-generational protocol whereby germ-line KO lines were first generated individually for a sample of smORFs, before their phenotype was scored in their fully mutant offspring. Their observed KO mutant phenotype rate of 37% approaches our own rate of 45.6% (including several coincident hits) and suggests that, at least for smORFs, full mutants must be used for genetic screenings.

Crossing mutant alleles to a deficiency has long been used as a test to classify mutant alleles by comparing the hemizygous phenotype with the homozygous allele phenotype [58]. The logic (and empirical observation) is that, as the deficiency is a null allele of the gene, hemizygosity would produce a greater loss of function that homozygosity for a partial loss of function allele. In our case, the increase in loss of function allowed for the use of publicly available insertional alleles that may not be null and increased the chance of revealing a mutant phenotype. Even in the case of publicly available null alleles, a second benefit of the use of a hemizygous test is that the confounding phenotypic effects of the modifiers accumulated in balanced stocks of null allele-carrying chromosomes are avoided. Our tests were carried out with the intention to identify promising candidates for a detailed characterisation (see below) rather than to provide an exhaustive genome-wide assessment of smORF function; however, our method could be scaled up, not only in *Drosophila* but also in other model organisms where mutant alleles are publicly available or easily generated.

Our screening and that of Bosch et al., 2023 [34] are largely complementary in the targeted genes; therefore, combining our results with those generated by the Bosch et al., 2023 [34] KO lines and successful somatic lines provides a reasonable sample (some 8.5% of the total sCDS) for a genome-wide assessment of smORF function in *Drosophila*. Such a combination would increase the number of sCDS showing a mutant phenotype to 62% of the tested sCDSs (Table 2). This is a number in line with the observations in yeast and bacteria [24,25,52] and is hence likely closer to the real fraction of smORFs producing biologically active peptides and microproteins. This is an important conclusion for smORF and genome studies; however, in the case of metazoans, it might only be applicable to sCDSs or to smORFs with a corroborated translation. However, the results for *Drosophila*, as well as the yeast data [24,52], indicate that screening cultures in different media or under different conditions is an important way to reveal smORF function: Bosch et al., 2023, [34] identified function in 10 smORFs raised under starvation or other restrictive media, whereas here the function of CG12384/*dap1* as a defining member of a new gene family of TOR-related negative regulators of autophagy [56] was only revealed under starvation.

## Figures and Tables

**Figure 1 cells-13-02090-f001:**
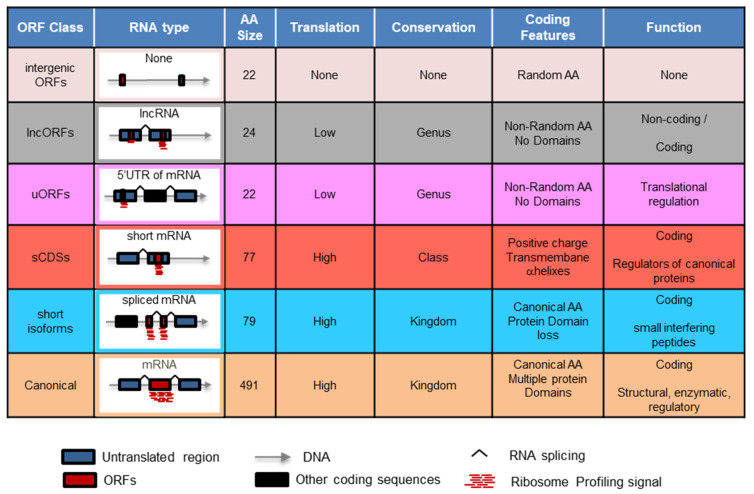
Types of smORFs. Types of ORFs mentioned in this work, and their relevant characteristics.

**Figure 2 cells-13-02090-f002:**
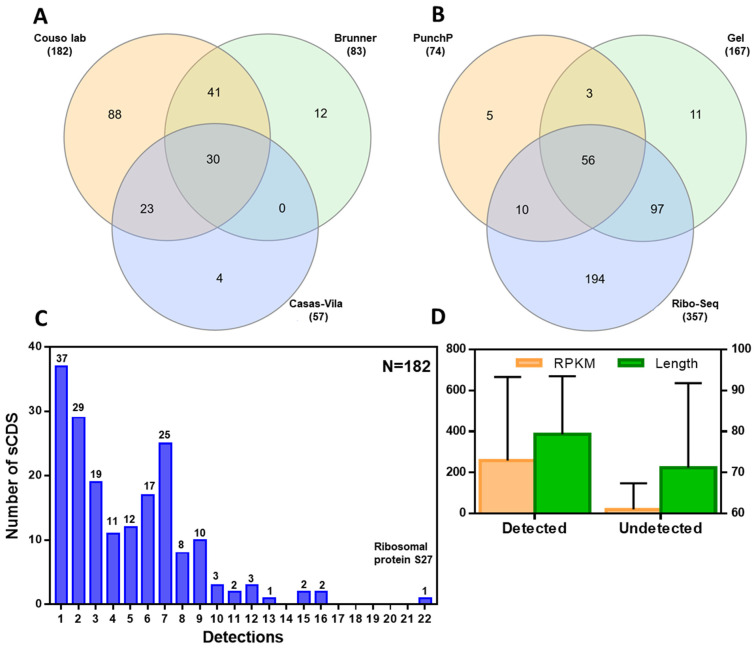
Proteomic detection of sCDS peptides. (**A**) Number of sCDS peptides and microproteins detected in this study (Couso lab) compared with previous data from Brunner and Casas-Vila [35,49]. (**B**) sCDSs detected by either PunchP or gel fractionation compared with those detected as translated by Ribo-seq [36]. (**C**) Proteomic hits amongst detected sCDSs, showing the number of times sCDSs were detected either in different experiments or by different peptides. (**D**) Different Ribo-seq RPKM levels [36] and length (aa) between sCDSs detected by proteomics (Couso lab) and those that remained undetected.

**Figure 3 cells-13-02090-f003:**
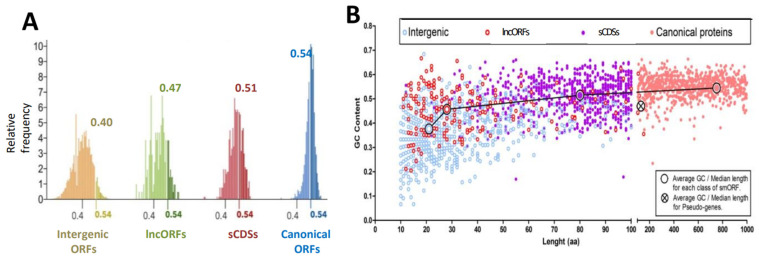
GC content of sCDSs and other ORFs. (**A**) Size distribution of the GC ratio for different ORF classes. Averages are shown on top and the average canonical value (0.54) is shown for the reference in each class. Only 5% of lncORFs score higher than this value, whereas almost half of sCDSs do. (**B**) The GC ratio (see methods) increases in apparent correlation with ORF type and length until values similar to those of canonical ORFs, and superior to those of pseudogenes, are reached by sCDSs. Large circles joined by a line represent the averages for each smORF class, pseudogenes, and canonical protein coding genes. See Appendix A for the data.

**Figure 4 cells-13-02090-f004:**
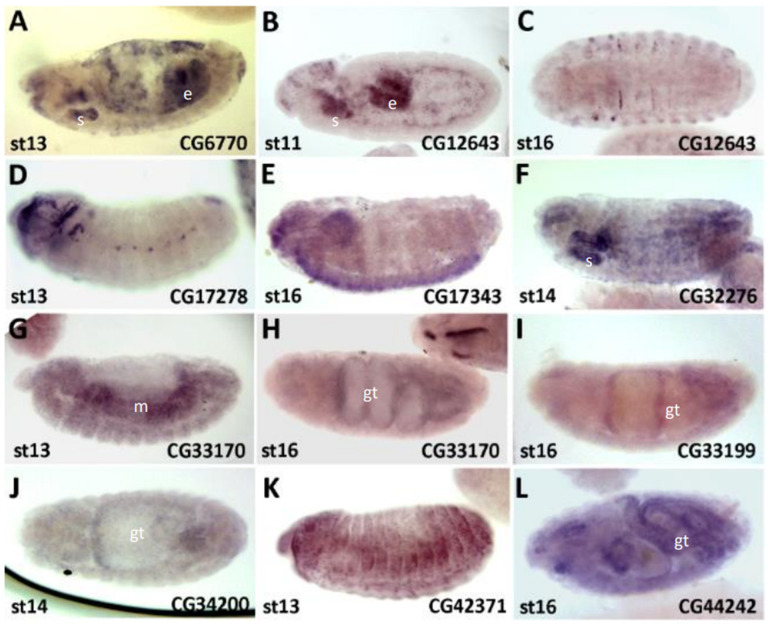
Patterns of sCDS expression. (**A**–**L**) Patterns of sCDS expression revealed by *in situ* hybridisation in *Drosophila* embryos, anterior to the left and dorsal-up; embryonic stage (st) is indicated on left bottom corner of each panel. Most patterns appear to first be located in the developing mesodermal (m) and endodermal (e) tissues, and then to their derivatives such as the gut (gt) and salivary glands (s). *CG17278* and *CG17343* are expressed in ectodermal tissues such as sensory organ precursors, and CNS, respectively. Most other patterns were either ubiquitous or faint (see Appendix A for further details).

**Figure 5 cells-13-02090-f005:**
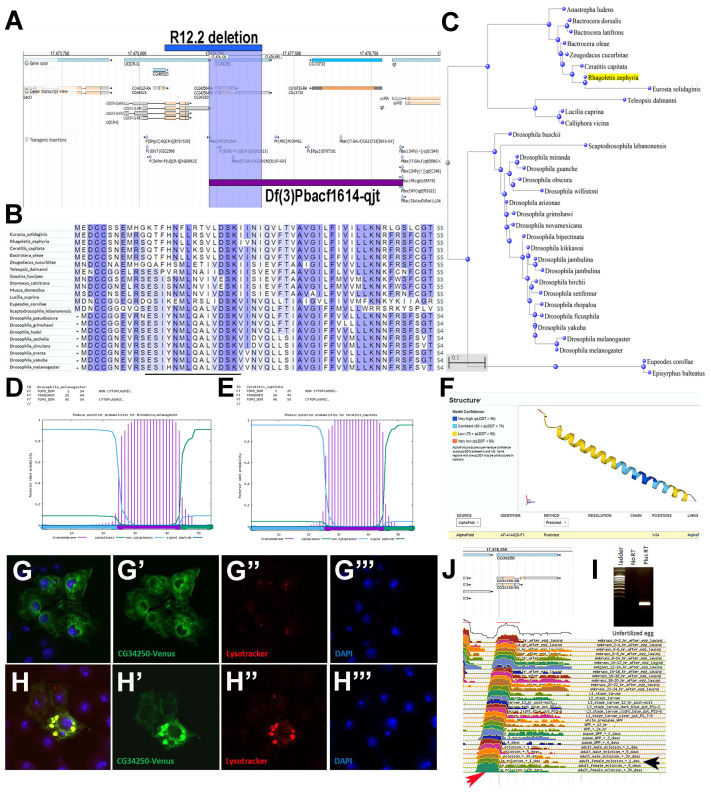
Functional characterisation of *CG34250*. (**A**) Flybase Drosophila genome browser displaying the *CG34250* locus and R12.2- and Df(3)f1614-qjt-generated deletions (blue and purple lines, respectively). (**B**) Aminoacid sequence alignment of CG34250 smORF family members, showing sequence similarities (blue); the MS-detected peptide is highlighted with a black bar. (**C**) Neighbour-Joining phylogenetic tree using Constraint-Based Multiple Alignment (COBALT) showing the evolutionary distances of CG34250 homologues. (**D**,**E**) Panels show the secondary structure predictions of signal peptide and transmembrane domains in CG34250 peptides obtained via the Phobius program from *Drosophila melanogaster* (**D**) and *Ceratitis capitata* (**E**), showing a conserved single transmembrane topology (bars). (**F**) 3D predicted structure of *Drosophila* CG34250 peptide using the Alphafold program, displaying a helical structure (**G**–**G’’’**). The transfection and expression of the CG34250 peptide tagged with Venus reveal initial reticular expression in the cytoplasm (**G’**, green), possibly in the ER. Lysotracker stains the lysosomes (**G’’**, red) and DAPI stains the nuclear DNA (**G’’’**, blue) (**H**–**H’’’**). The CG34250 peptide tagged with Venus (**H’**) accumulates at lysosomes (**H’’**, red); DAPI stains the nuclei (**H’’’**). (**J**) Flybase genome browser showing *CG34250* RNA expression levels at all stages studied (red arrow), including adult mature females (black arrow). (**I**) RT-PCR amplification of *CG34250* transcript fragment (195 bp) from messenger RNA extracted from unfertilized eggs.

**Figure 6 cells-13-02090-f006:**
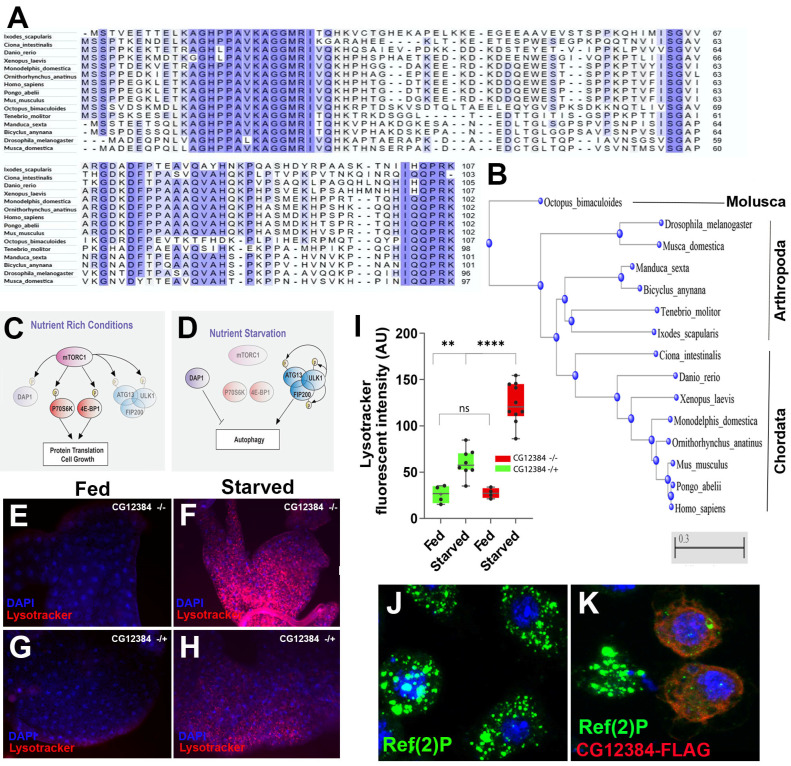
Functional characterization of CG12384/*Drosophila* DAP. (**A**) Aminoacid sequence alignment of death-associated protein 1 family members, with similarities shown in blue. (**B**) Neighbour-Joining phylogenetic tree using Constraint-Based Multiple Alignment (COBALT), showing that CG12384 peptides belong to a highly conserved smORF family with the members present in Deuterostomes (Chordata) and Protostomes (Arthropoda, Molusca). (**C**) The suggested role of DAP1 in mouse TOR pathway (Koren et al., 2010) [56]. Under normal conditions, mTORC1 is proposed to repress DAP1 function (mediated by phosphorylation). (**D**) Under nutrient restriction, the repression of mTORC1 allows DAP1 to repress excessive autophagy (autophagy is an initial beneficial cellular response but, if maintained, produces apoptosis). (**E**–**H**) The increase in lysosomes in the *Drosophila* fat body cells of starved larvae is much more intense in *CG12384* mutants (**F**) than in the wild-type (**H**); note that minimal increase in lysosomes occur during fed conditions in both *CG12384* mutant (**E**) and wild-type fat body cells (**G**). (**I**) Quantification of lysotracker fluorescence as shown in (**E**–**H**), showing the drastic increase in lysosomal activity in *CG12384* mutants during starvation conditions (****: *p* < 0.0001, **: *p* = 0.0024, ns: non-significant, as obtained by standard two-tailed unpaired *t*-test). (**J**,**K**) S2 cells grown in serum-free medium show an upregulation of the autophagosome marker Ref(2)P, but this increase is prevented in cells overexpressing a CG12384-Flag peptide. Note also that Flag and Ref(2)P seem not to overlap. The antagonism between Ref(2)P and CG12384-Flag is compatible with the proposed role for DAP1 in the upstream regulation of the autophagy response downstream of TOR.

**Table 1 cells-13-02090-t001:** Genetic assessment of sCDS function.

Gene Name	Hemizigous Phenotype	Relative Fitness	Overall Phenotype
CG11985	lethal		mutant
CG12384	lethal		mutant
CG13018	lethal		mutant
CG14036	low viability	lower fitness	mutant
CG17059	low viability		mutant
CG17278	low viability, leg phenotype		mutant
CG17343	lethal		mutant
CG17931	low viability		mutant
CG30415	lethal		mutant
CG33713	low viability		mutant
CG33714	semi-lethal		mutant
CG34250	viable	lower fitness	mutant
CG42497	lethal		mutant
CG44242	semi-lethal		mutant
CG6610	lethal		mutant
CG7630	viable	lower fitness	mutant
CG7646	viable, 50% heldout wings	equilibrated	mutant
CG8498	low viability		mutant
CG8788	viable	lower fitness	mutant
CG9034	lethal		mutant
l(2)06225	lethal		mutant
CG10418	viable	higher fitness	wild-type
CG12643	viable		wild-type
CG12994	viable		wild-type
CG13751	viable		wild-type
CG14104	viable	higher fitness	wild-type
CG15386	viable		wild-type
CG17127	viable		wild-type
CG17776	viable		wild-type
CG18081	viable	higher fitness	wild-type
CG18622	viable		wild-type
CG32267	viable	higher fitness	wild-type
CG32276	viable	higher fitness	wild-type
CG32368	viable		wild-type
CG33169	viable	higher fitness	wild-type
CG33170	viable	higher fitness	wild-type
CG3321	viable	higher fitness	wild-type
CG33672	viable	equilibrated	wild-type
CG34200	viable		wild-type
CG34242	viable	higher fitness	wild-type
CG42371	viable		wild-type
CG42394	viable	higher fitness	wild-type
CG5446	viable		wild-type
CG6615	viable		wild-type
CG6770	viable	higher fitness	wild-type
CG8860	viable		wild-type

**Table 2 cells-13-02090-t002:** Functional indicators in *Drosophila* sCDSs. We collected data for the sCDSs that were genetically assessed in this work and in that of Bosch et al., 2023 (top two rows), and also for the sCDSs studied by Patraquim et al., 2020 [36] (bottom row). We calculated the percentage with evidence of proteomics according to our results and those of Brunner et al., 2007 [49] or Casas-Vila et al., 2017 [35] (Figure 2); GC content; average length; percentage showing Drosophilid and human conservation; and the percentage for which a phenotype was obtained in other genetic tests, such as RNAi screens in S2 cell cultures [32]. * For comparison, the percentage of transcribed sCDSs for which embryonic translation was observed via Ribo-seq is 70.7% [36]. ^+^ Note that the data for human and Drosophilid conservation in genetically assessed sCDSs are skewed by the fact that Bosch et al., 2023 [34] only screened human-conserved smORFs, and are presented here to compare the sCDSs showing mutant (top row) or viable phenotypes (middle row).^@^: these data are extrapolated from the combination of this work and that of Bosch et al., 2023 [34].

Drosophila sCDSs	Proteomic Detection	GC Content	Length (aa)	Drosophilid Conservation	Human Conservation	RNAi Hits (Flybase)	Lower Mutant Viability	Number of sCDS
Averages and % for lethal or lowly viable	62%	0.54	80.9	75% ^+^	68% ^+^	50%	100%	47
Averages and % for fully viable	55%	0.53	76.8	65% ^+^	48% ^+^	35%	0%	29
Averages and % for all sCDSs	55.4% *	0.51	72.8	47.5%	34.5%	27.5%	62% ^@^	862

## Data Availability

The relevant data for proteomics can be found at https://github.com/CousoLab/Cells_smORFs_peptides_repository.git (accessed on 8 December 2024).

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
