# Peer review of "Pervasiveness of Microprotein Function Amongst Drosophila Small Open Reading Frames (SMORFS)"

_cells, 2024, doi:10.3390/cells13242090_

Round 1

Reviewer 1 Report

Comments and Suggestions for Authors

The stated goal of this study is to detect and corroborate microprotein function in Drosophila. This is a challenge and so as far as the motivation of this work is concerned, it is sound and worth pursuing. However I had trouble clearly understanding the analyses and experiments that the authors performed and exactly how they claim that these support their conclusions. The structure and writing of the manuscript was not as clear as it needs to be, and the image quality and presentation of the figures was problematic. Additionally, there are important methodological details that relate to the bioinformatics analyses that are entirely missing. Reading this manuscript, one detects a lack of attention; for example in the introduction, numbers and percentages of sORFs are presented, without any mention to which organism they correspond (we get to Drosophila later). 

A fundamental problem is that the authors repeatedly conflate proteomics detection with function (l. 109, 225, 266 and more). This is simply not true, and while detection by MS-based proteomics is more reliable than detection by Ribo-seq experiments, one cannot claim that the detected microproteins are functional without further experiments. The authors go on to test mutants for a subset of 43 out of a total of 182 sCDS and only find phenotypic effects in 20 of them. Smaller subsets even are the focus of additional analyses, and two are chosen for further validation. One of them is found to have a transmembrane domain and to have a cellular localization pattern consistent with a membrane protein. And as far as I can understand, this is the only evidence relating these microproteins to membranes, yet in the abstract the authors claim that their results support a regulatory role of membrane proteins.

Another critical issue is that the authors base much of their results on the claim that high GC is linked with coding potential, because known coding regions have higher GC than noncoding ones. Here a proper analysis is not presented, and we are only given a comparison among categories (how were these defined? there are no methods) and the conclusion that because sCDS have average GC that is closer to CDS than lncORFs do, they are protein-coding. But this does not follow as simple as that, and the fact that higher GC leads to a higher chance of obtaining a longer ORF by chance, simply due to the fact that stop codons are AT-rich, is not discussed. 

The analysis of sequence conservation unfortunately also suffers from the lack of methodological details and general clarity. Exactly what data and from which species were used for the similarity searches? What criteria were used? Perhaps I missed it but I do not see where the number of 887 sCDS comes from (l 318). 

Comments on the Quality of English Language

None

Author Response

Comments and Suggestions for Authors

“The stated goal of this study is to detect and corroborate microprotein function in Drosophila. This is a challenge and so as far as the motivation of this work is concerned, it is sound and worth pursuing. However I had trouble clearly understanding the analyses and experiments that the authors performed and exactly how they claim that these support their conclusions. The structure and writing of the manuscript was not as clear as it needs to be, and the image quality and presentation of the figures was problematic. Additionally, there are important methodological details that relate to the bioinformatics analyses that are entirely missing. Reading this manuscript, one detects a lack of attention; for example in the introduction, numbers and percentages of sORFs are presented, without any mention to which organism they correspond (we get to Drosophila later).”

The whole manuscript has undergone a thorough review, with the addition of many data, both in the manuscript and as supplementary data. For some reason, some reviewers did not have access to the previously submitted supplementary data, where some of the requested information was to be found.

Here in the introduction, we have changed the text to clarify numbers for Drosophila and for humans.

“A fundamental problem is that the authors repeatedly conflate proteomics detection with function (l. 109, 225, 266 and more). This is simply not true, and while detection by MS-based proteomics is more reliable than detection by Ribo-seq experiments, one cannot claim that the detected microproteins are functional without further experiments. The authors go on to test mutants for a subset of 43 out of a total of 182 sCDS and only find phenotypic effects in 20 of them. Smaller subsets even are the focus of additional analyses, and two are chosen for further validation. One of them is found to have a transmembrane domain and to have a cellular localization pattern consistent with a membrane protein. And as far as I can understand, this is the only evidence relating these microproteins to membranes, yet in the abstract the authors claim that their results support a regulatory role of membrane proteins.”

We are always treating the different functional indicators as that, mere indicators, not as proof of functionality. We have corrected the language throughout the manuscript to make this clearer and introduced further sentences (i.e. see pages 4 and 21). We have also added mutant tests for another 4 sCDS, and updated Tables 1 and 2 accordingly. However, it has to be remembered that these Drosophila sCDS are about 80aa long, and are annotated as coding by the Fly genome Project. Here we are presenting new lines of evidence to corroborate this “coding hypothesis” (which we thought to be non-controversial), that could have been disputed by taking the results of Bosch et al. 2023 at face value. Finally, the literature indicates that, at least for smORFs, RiboSeq detection is more reliable than proteomics. The reviewer might mean that bona-fide proteomics detection is a more reliable indicator of function that RiboSeq, but that both still need experimental corroboration by genetics, and we are in agreement.

We also provide now further evidence for membrane-association, both as citations and further data.

“Another critical issue is that the authors base much of their results on the claim that high GC is linked with coding potential, because known coding regions have higher GC than noncoding ones. Here a proper analysis is not presented, and we are only given a comparison among categories (how were these defined? there are no methods) and the conclusion that because sCDS have average GC that is closer to CDS than lncORFs do, they are protein-coding. But this does not follow as simple as that, and the fact that higher GC leads to a higher chance of obtaining a longer ORF by chance, simply due to the fact that stop codons are AT-rich, is not discussed.”

We have included a supplemental file with the data for the GC analysis, and a new section in methods about this analysis. Sequences for different smORF classes were extracted from Flybase annotations and previous publications. The reviewer is correct that correlation between length and GC content has been reported (and can be observed in our data). However, the average values between putative non-coding sequences (such as lncORFs and pseudogenes, with a 0.47 GC ratio) and putative coding ones (such as canonical ORFs and sCDS, with GC ratios of 0.54 and 0.51 respectively) are different enough to be used as an indicator of possible coding function in combination with other markers (not a proof). GC-inflation by ATG/Stop codon-mediated AT-depletion in longer sequences does not seem an explanation for sCDS with a mutant phenotype, which have a maximum length of 99 aa and an average of 77aa, yet have an average GC ratio of 0.54, identical to canonical ORFs that have minimum lengths of 100aa and an average of near 500aa (see Fig. 1, Table 2 and Sup. file S2). This is supported by recent works such as Wang et al. 2019, Bioinformatics 35: 2949-2956, which also reports clear differences in GC content between lncRNAs and coding ORFs. This and other references that were in the discussion have now been introduced in the relevant section of the results.

“The analysis of sequence conservation unfortunately also suffers from the lack of methodological details and general clarity. Exactly what data and from which species were used for the similarity searches? What criteria were used? Perhaps I missed it but I do not see where the number of 887 sCDS comes from (l 318).”

There is a new section in methods about this analysis. The data for human-fly homologies was obtained from Flybase and EMBL annotations, and supplemented with data from Bosch et al. 2023. Drosophilid conservation was obtained from Flybase and Patraquim et al. 2022, and so is the number of sCDS, which actually is 862 (corrected).

Reviewer 2 Report

Comments and Suggestions for Authors

The authors in this report assessed Drosophila smORFS/sCDS available in data bases in order to get functional clues on these sequences as most of which, albeit considered as important informations, remain unknown for a number of reasons described in the text. 

The methodological approach combines a variety of bioinformatic, proteomic and genetical tools as well. I find an overall excellent manuscript containing a powerful set of techniques, very good/informative figures illustrating significant results as well. In addition, the data support the conclusions drawn by the authors and, moreover, expand the knowledge horizon of smORFS role as the report design allowed to increase the micro peptide identification.

Just minor points:

Given the title format of Cells, "SMORFS" may be read/sound strange or unknown to some readers. If lower case cannot be used in the title, I suggest replacing "SMORFS" by  SMALL OPEN READING FRAMES so that the concept will be readily recognised.

Lines 372, 373 and 406 display three references. Please check 372 and 373 whether they really appear in the reference list while replacing Bosch et al. 2023 by its corresponding number(33) cited in the list.

Author Response

Comments and Suggestions for Authors

The authors in this report assessed Drosophila smORFS/sCDS available in data bases in order to get functional clues on these sequences as most of which, albeit considered as important informations, remain unknown for a number of reasons described in the text.

The methodological approach combines a variety of bioinformatic, proteomic and genetical tools as well. I find an overall excellent manuscript containing a powerful set of techniques, very good/informative figures illustrating significant results as well. In addition, the data support the conclusions drawn by the authors and, moreover, expand the knowledge horizon of smORFS role as the report design allowed to increase the micro peptide identification.

Just minor points:

“Given the title format of Cells, "SMORFS" may be read/sound strange or unknown to some readers. If lower case cannot be used in the title, I suggest replacing "SMORFS" by  SMALL OPEN READING FRAMES so that the concept will be readily recognised.”

We have changed the title accordingly

“Lines 372, 373 and 406 display three references. Please check 372 and 373 whether they really appear in the reference list while replacing Bosch et al. 2023 by its corresponding number(33) cited in the list.”

Done.

Reviewer 3 Report

Comments and Suggestions for Authors

Proteins from smORFs are difficult to detect due to the smaller RNA/protein size and lower abundance, and the function are not yet clear because previous Riboseq method cannot track the fate or function after translation. In this study, Platero et al. utilized a proteomic method to discover functional microproteins from smORF in Drosophila. They focused on sCDS proteins at 10kDa and validated the function of several sCDS proteins. The manuscript is well written, and the discovery is novel. I only have a few suggestions/requirements about the integrity of this manuscript before considering a publication.

1. First, the figure number starts from Fig.2, and all supplementary figures are not found. The authors should check their submission.

2. The design of their mass spec experiment is not very well elaborated, and the result is not displayed appropriately. The authors must provide the details to allow audiences to evaluate the quality of this dataset.

2.1        What is the control in this case? At least the authors should run mass spec on the blank gel at the corresponding location.

2.2        How many replicates did they do? It appears to this reviewer that they only have one replicate.

2.3        The authors must show their original mass spec results, including intensity, unique peptide of all experimental conditions and control conditions.

3. The authors omitted many data in the session of “Validation of genetic results”

Line 433: “which we corroborated by RT-PCR of unfertilized eggs derived from CG34250 females.” No data is shown to support this claim.

Line 436: “Surprisingly, we observed no effect until the crosses were done in such a way that the CG34250 null progeny from CG34250 null mothers were raised in competition with non-mutant siblings.” What is the phenotype here? The authors must include data to support their conclusion.

Line 477: “This insertion allowed us to recover homozygous mutant larvae, in which we observed increased autophagy under starvation, whereas over-expression of this peptide prevents this effect.” Again, there is no data to support this claim.

Author Response

Comments and Suggestions for Authors

“Proteins from smORFs are difficult to detect due to the smaller RNA/protein size and lower abundance, and the function are not yet clear because previous Riboseq method cannot track the fate or function after translation. In this study, Platero et al. utilized a proteomic method to discover functional microproteins from smORF in Drosophila. They focused on sCDS proteins at 10kDa and validated the function of several sCDS proteins. The manuscript is well written, and the discovery is novel. I only have a few suggestions/requirements about the integrity of this manuscript before considering a publication.

  1. First, the figure number starts from Fig.2, and all supplementary figures are not found. The authors should check their submission.”

We have checked that figure calls start at 1. As mentioned in the response to reviewer 1, some reviewers seem not to have access to some supplementary files. We have now included all supplementary data as a single zip.

“2. The design of their mass spec experiment is not very well elaborated, and the result is not displayed appropriately. The authors must provide the details to allow audiences to evaluate the quality of this dataset.”

We have added details to materials and methods, and we have incorporated two new figures and panels (figure 2D and S2), plus a supplementary file (S1) with proteomic information.

“2.1        What is the control in this case? At least the authors should run mass spec on the blank gel at the corresponding location.”

We have carried out the control experiment as proposed by the reviewer. We have analyzed by LCMS an empty gel band corresponding to the size of our proteins of interest, aprox. 10KDa.

As a result we have obtained 14 peptides, but only two correspond to sCDS. These two peptides are LIAAQTGTK (FBpp0085529) and VVLNKLKK (FBpp0312014).

We have removed the LIAAQTGTK peptide from our experimental proteomics results (supplementary file 1) but the sCDS CG3450 corresponding to FBpp0085529 has not been removed, since we have other peptide identifications for it. The VVLNKLKK peptide was not on our previous list nor is the sCDS and thus it does not affect our results.

“2.2        How many replicates did they do? It appears to this reviewer that they only have one replicate.”

As indicated in supplementary file 1 and Supplemental Figure S1B, we have carried out 36 proteomic extractions (including 29 replicates of the PunchP experiment and 7 of the Gel extraction), plus the control.

2.3        The authors must show their original mass spec results, including intensity, unique peptide of all experimental conditions and control conditions.

We have incorporated a supplementary file S1 with proteomics information and also two new figures (Figure 2D and S2) including the spectra for the two sCDSs that are characterised in more depth (CG34250 and CG12384).

“3. The authors omitted many data in the session of “Validation of genetic results”

Line 433: “which we corroborated by RT-PCR of unfertilized eggs derived from CG34250 females.” No data is shown to support this claim.

We have incorporated a new panel in Figure 5I.

Line 436: “Surprisingly, we observed no effect until the crosses were done in such a way that the CG34250 null progeny from CG34250 null mothers were raised in competition with non-mutant siblings.” What is the phenotype here? The authors must include data to support their conclusion.

The phenotype is lower viability. The data was available in the previous Sup. Table 1, and now is expanded and available in supplemental file 4.

Line 477: “This insertion allowed us to recover homozygous mutant larvae, in which we observed increased autophagy under starvation, whereas over-expression of this peptide prevents this effect.” Again, there is no data to support this claim.

We have incorporated new panels in Figure 6 E,F,G,H and I, including a quantification of lysotracker fluorescence changes.

Round 2

Reviewer 3 Report

Comments and Suggestions for Authors

The authors addressed all of my questions. Good job!

Author Response

Thank you